# Can Pre-trained Vision and Language Models Answer Visual Information-Seeking Questions?

**Yang Chen**[♠♡*]   **Hexiang Hu**[♠]   **Yi Luan**[♠]   **Haitian Sun**[♠]   **Soravit Changpinyo**[♣]
**Alan Ritter**[♡]   **Ming-Wei Chang**[♠]
[♠]Google Deepmind   [♣]Google Research   [♡]Georgia Institute of Technology

## Abstract

Pre-trained vision and language models (Chen et al., 2023b,a; Dai et al., 2023; Li et al., 2023b) have demonstrated state-of-the-art capabilities over existing tasks involving images and texts, including visual question answering. However, it remains unclear whether these models possess the capability to answer questions that are not only querying visual content but knowledge-intensive and information-seeking. In this study, we introduce INFOSEEK[1], a visual question answering dataset tailored for information-seeking questions that cannot be answered with only common sense knowledge. Using INFOSEEK, we analyze various pre-trained visual question answering models and gain insights into their characteristics. Our findings reveal that state-of-the-art pre-trained multi-modal models (e.g., PaLI-X, BLIP2, etc.) face challenges in answering visual information-seeking questions, but fine-tuning on the INFOSEEK dataset elicits models to use fine-grained knowledge that was learned during their pre-training. Furthermore, we show that accurate visual entity recognition can be used to improve performance on INFOSEEK by retrieving relevant documents, showing a significant space for improvement.

## 1   Introduction

The acquisition of knowledge occurs in the pre-training of large language models (Brown et al., 2020; Chowdhery et al., 2022), demonstrated as their emergent ability to answer information-seeking questions in the open-world, where the questioner does not have easy access to the information. While prior works have analyzed models' capabilities to answer *textual* information-seeking (or info-seeking) questions, much less is known for *visual* info-seeking questions. For example,

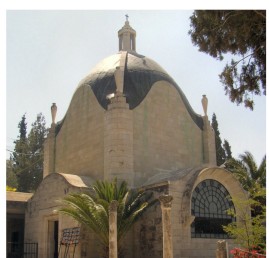

Figure 1: While 70.8% of OK-VQA questions can be answered by average adults without using a search engine, INFOSEEK poses challenges to query fine-grained information about the visual entity (e.g., `Dominus Flevit Church`), resulting in a sharp drop to 4.4% (§2).

after taking a picture of the specific church in Figure 1, a person might want to know the date of construction, or who decorated the interior of the church. Although the entity is presented in the image (the specific church), the relevant knowledge (e.g., the date) is not. Given recent advances on pre-trained visual and language models (Alayrac et al., 2022; Chen et al., 2023b; Li et al., 2023b), *do these models also understand how to answer visual information-seeking questions?*

To study this research question, a visual question answering (VQA) dataset focusing on info-seeking questions is inevitably required. However, not all VQA datasets meet this criterion. For example, by design, the majority of questions in datasets such as VQA v2 (Goyal et al., 2017) focus on visual attributes and object detection that does not require information beyond the image to answer. While models capable of answering these types of questions have the potential to aid visually impaired individuals (Gurari et al., 2018), there is a broader class of *info-seeking* questions that cannot be easily answered by sighted adults. Handling such questions (e.g., `When was this building constructed? 1955`) is critical as they come closer to the natural distribution of human questions.

In this paper, we present INFOSEEK, a natural

---

[*] Work done when interned at Google
[1]Our dataset is available at https://open-vision-language.github.io/infoseek/.

VQA dataset that focuses on visual info-seeking questions. Different from previous VQA datasets, the testing subset of INFOSEEK is collected in multiple stages from human annotators to evaluate VQA where the question can not be answered from only the visual content (see a comparison of datasets in § 2). In addition to this manually curated test set, which enables realistic evaluation of info-seeking VQA, we also join annotations from a recent visual entity recognition dataset (Hu et al., 2023) with the Wikidata database (Vrandečić and Krötzsch, 2014), and employ human annotators to write templates to semi-automatically generate a large corpus of visual info-seeking QA pairs. Over 1 million {image, question, answer} triplets are generated to support fine-tuning multimodal models for info-seeking VQA. We split data to ensure memorizing knowledge during fine-tuning is useless — models either have to learn to use knowledge learned during pre-training or learn to retrieve knowledge from an external knowledge base.

Using INFOSEEK, we analyze the ability of state-of-the-art models to answer visual info-seeking questions. We found pre-trained vision-language models, such as models pre-trained end-to-end (e.g., PaLI-X by Chen et al.), and models pre-trained with frozen LLM (e.g., BLIP2 by Li et al.), both struggle to answer info-seeking questions in zero-shot, though BLIP2 outperforms PaLI-X by a margin. Surprisingly, after fine-tuning on our (large, semi-automatically curated) training set, PaLI-X yields a significant improvement and outperforms the fine-tuned BLIP2 models on queries that are unseen during fine-tuning. This suggests that while pre-trained PaLI-X has a significant amount of knowledge, it requires a small amount of fine-tuning data to fully awaken its capabilities. Furthermore, we show that INFOSEEK fine-tuned models can even generalize to questions and entity types completely unseen during fine-tuning (e.g., art & fashion).

When incorporating a visual entity recognition component, and conditioning models on the Wikipedia articles of the relevant entities, we show that models accessing such a knowledge base (With-KB) perform better overall than those that rely on knowledge learned during pre-training. However, end-to-end (No-KB) models were found better on certain classes of questions that require coarse-grained answers (*"Which continent is this building located on?"*), even on tail entities. Our

| Dataset | OK-VQA | ViQuAE | INFOSEEK |
|---|---|---|---|
| PaLM (Q-only) | 23.8 | **31.5** | 5.6 |
| Current SotA | 66.1 | 22.1 | 18.2 |
| Require Knowledge[†] | 29.2% | 95.2% | 95.6% |

† :% of questions that require knowledge to answer.
PaLM (Q-only): a question-only baseline using PaLM.

Table 1: Comparison of INFOSEEK and prior KI-VQA benchmarks. Performances reported in VQA score.

experiment (§5.2) further suggests that improving visual entity recognition can drastically increase model's capability in answering visual info-seeking questions (from 18% to 45.6%), indicating a promising direction for future development.

## 2 The Need for a New Visual Information-seeking Benchmark

While there have been plenty of knowledge-intensive VQA (KI-VQA) benchmarks, we show that none of these meet the criteria to effectively evaluate info-seeking VQA. Early efforts in this area, such as KBQA (Wang et al., 2015) and FVQA (Wang et al., 2017), were based on domain-specific knowledge graphs, while recent datasets like OK-VQA (Marino et al., 2019) and its variants such as S3VQA (Jain et al., 2021) and A-OKVQA (Schwenk et al., 2022) have improved upon this foundation by incorporating an open-domain approach and highlighting common-sense knowledge. Among the existing benchmarks, K-VQA (Sanket Shah and Talukdar, 2019) and ViQuAE (Lerner et al., 2022) are the most relevant, but they have severe limitations in their question generation process, as discussed below.

**Information Seeking Intent.** The evaluation of models' ability to answer info-seeking questions requires fine-grained knowledge, which a person is unlikely to know off the top of their head. However, we found that 70.8% of OK-VQA questions[2] can be answered without the need to use a search engine, indicating the dataset primarily focuses on knowledge that is commonly known to people. Most OK-VQA questions are regarding coarse-grained knowledge that many people already know: `What days might I most commonly go to this building? Sunday`. One only needs to know the building type (e.g., `Church`) rather than the specific building (e.g., `Dominus Flevit Church`). This

[2]Studied with human on 500 random OK-VQA questions (see Appendix C.1)

makes it unsuitable for evaluating pre-trained models on long-tailed knowledge, where these models have shown weaknesses (Kandpal et al., 2022).

**Reliance on Visual Understanding.** In contrast to OK-VQA, the ViQuAE dataset aims to test fine-grained knowledge of visual entities by pairing questions from TriviaQA (Joshi et al., 2017) with images. However, a significant portion of the ViQuAE questions (e.g., "Who betrayed him for 30 pieces of silver?") can be answered without looking at the images, as the questions often reveal sufficient information to determine the answer. To quantify this observation, we present questions from the evaluation set to a large language model, PaLM (540B) (Chowdhery et al., 2022). Results on the ViQuAE test set are shown in Table 1. Surprisingly, we find that PaLM can read questions and generate answers with 31.5% accuracy, outperforming the SOTA retrieval-based model (Lerner et al., 2022) (which has access to the image) on this dataset by 9.4%. Although PaLM is a much larger model, this experiment illustrates that it is possible to achieve very good performance on ViQuAE without using information from the image.

**Entity Coverage.** Current VQA datasets often cover a limited number of visual entity categories. For example, K-VQA only focuses on human subjects, while over 43% of questions in ViQuAE revolve around human entities (see Table 2). Such limitations hinder the evaluation of a model's knowledge across various entity categories and may result in reduced task complexity, as the evaluation may be limited to mere facial recognition.

To address these limitations, we present INFOSEEK (§ 3), a new benchmark for pre-trained multimodal models on visual info-seeking questions. Our work builds on top of a visual entity recognition dataset, OVEN (Hu et al., 2023), which is designed to answer questions related to the identification of visual entities. We take visual info-seeking a step further by benchmarking info-seeking questions about visual entities, which allows us to test the pre-training knowledge of models beyond simply recognizing an entity.

## 3 INFOSEEK: A VQA Benchmark of Visual Information-seeking Questions

The INFOSEEK dataset consists of two components, (1) INFOSEEK $_{Human}$: a collection of human-written visual info-seeking questions (8.9K) to simulate information seeking intent (see § 3.1); and (2)

| Dataset | # {I, Q, A} | Len of Q/A | # Entity | # Ent. type |
|---|---|---|---|---|
| OK-VQA | 14K | 8.1/1.3 | - | -⋆ |
| K-VQA | 183K | 10.1/1.6 | 18,880 | 1† |
| ViQuAE | 3.6K | 12.4/1.7 | 2,397 | 980 |
| INFOSEEK | | | | |
| - Wikidata | 1.35M | 8.9/1.5 | 11,481 | 2,739 |
| - Human | 8.9K | 7.8/2.3 | 806 | 527 |

⋆: OK-VQA does not specify visual entities.
†: K-VQA only covers entities from the human category.

Table 2: Statistics of INFOSEEK & KI-VQA datasets.

INFOSEEK $_{Wikidata}$: an automated dataset (1.3M) to cover diverse entities for large-scale training and evaluation purposes (see § 3.2). We split the dataset to ensure memorizing the training set is useless, thereby emphasizing the importance of pre-training to acquire knowledge (see § 3.3). Due to space limitations, we summarize the key essence in this section and defer details to the Appendix.

**Image Sources for Diverse Entity Coverage.** We sourced images from 9 image classification and retrieval datasets used in Hu et al., including landmarks (17%), animals (13%), food (5%), aircraft (3%), etc. We utilize their annotation, which links visual entities to their corresponding Wikipedia articles, to construct our INFOSEEK dataset.

### 3.1 INFOSEEK$_{Human}$: Natural Info-Seeking VQA Data Annotated by Humans

To ensure INFOSEEK questions *rely on visual understanding* and prevent models from taking shortcuts in the question without using the image, we employ a two-stage annotation approach inspired by TyDiQA (Clark et al., 2020). This makes it unlikely questioners will have prior knowledge of the answer like SQuAD (Rajpurkar et al., 2016), ensuring questions with *info-seeking intents* (Lee et al., 2019).

**Question Writing.** Annotators are asked to write 3-5 questions about a visual entity based on their own curiosity and information needs. To aid the question-writing process, they are prompted with visual entity images, a short description (15 words) about the entity, and a list of Wikipedia section titles. This ensures that the questions reflect a genuine interest in learning about important aspects of the entity without seeing the answer. A set of annotation rules is employed to prevent trivial questions, such as questions about visual attributes.

**Answer Labeling.** For each entity, we randomly assign collected questions to a different group of an-

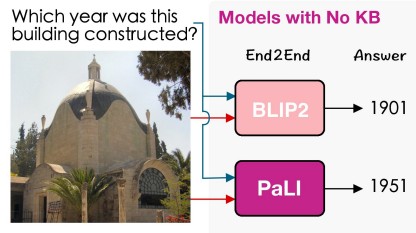

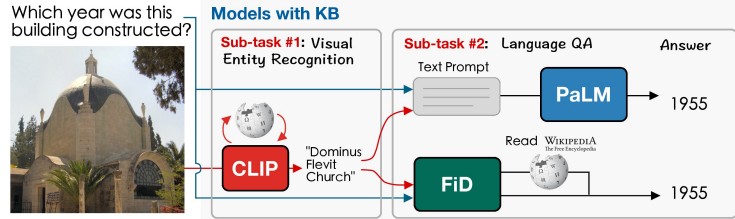

(a) Models with **No KB** access

(b) Models **With KB** (*Knowledge-base*) information

Figure 2: **Visual info-seeking models** under the proposed **No KB** and **With KB** protocols. (a) End-to-end VQA models (such as PaLI (Chen et al., 2023b) or BLIP2 (Li et al., 2023b)) that directly predict the answer from looking at the image and question; (b) Pipeline systems with access to a knowledge base (e.g.Wikipedia), with the option to link the queried subject to the Wikipedia use CLIP (Radford et al., 2021) and perform textual question-answering using PaLM (Chowdhery et al., 2022) or Fusion-in Decoder (FiD) (Izacard and Grave, 2020).

notators to label answers from Wikipedia. Annotators were shown the Wikipedia article of the entity and asked to find a concise answer to the question: a text span that is as short as possible while still forming a satisfactory answer. In addition, annotators categorize questions into three types: TIME (e.g., year), NUMERICAL (e.g., height) and STRING (e.g., location).

Finally, we construct {image, question, answer} (IQA) triples by assigning images for the annotated QA pair of a visual entity, followed by human verification and clarification of the questions if multiple objects are presented in the image. Following TyDiQA (Clark et al., 2020), we measure the *correctness* of annotations and take the high accuracy (95%) as evidence that the quality of the dataset is reliable for evaluating visual info-seeking models.

## 3.2 INFOSEEK_Wikidata: 1 Million Automated VQA Data from Wikipedia

Human annotation is valuable but costly for large-scale evaluation. We thus scale up the dataset using a semi-automated procedure, transforming knowledge triples in Wikidata (2022-10-03) to natural language questions with human-authored templates, resulting in 1.3M examples over 11K visual entities covering 2.7K entity types (see Table 2).

**QA Generation.** We convert knowledge triples (subj, relation, obj) in Wikidata to natural language question-answer pairs for a selected list of 300 relations. For each relation, annotators write one or two question templates, which contain a placeholder for a hypernym of the visual entity (e.g., car) and a placeholder for unit measurements (e.g., inches) in numerical questions to avoid ambiguity. Finally, we construct the IQA triples by pairing images of a visual entity with corresponding QA

pairs.[3]

**QA Pair Filtering and Subsampling.** To ensure the questions are diverse and the answers can be referenced from Wikipedia, we filter out QA pairs when answers from Wikidata cannot be found in the Wikipedia article and subsample questions to balance the distribution of entities and relations.

## 3.3 Evaluation of INFOSEEK

**Dataset Split.** We design the evaluation split to prevent overfitting to the training set and focus on evaluating the generalization ability of the pre-trained models. This includes the ability to answer questions of new entities and questions not seen during training. Particularly, we define two evaluation splits: (1) UNSEEN ENTITY, where a portion of entities are held out during training and only included in the evaluation; (2) UNSEEN QUESTION, where we hold out a portion of the QA pairs of seen entities for evaluation.

**Evaluation Metric.** Three types of questions are evaluated differently: VQA accuracy (Goyal et al., 2017) for STRING and TIME; *Relaxed Accuracy* (Methani et al., 2020) for NUMERICAL. We applied different relaxing strategies for each question type, and averaged the accuracy for each question. Finally, we calculate the accuracy for each data split (UNSEEN QUESTION and UNSEEN ENTITY), and take the harmonic mean of them as the overall accuracy (see Appendix).

## 4 Protocols and Models for INFOSEEK

Motivated by previous research on text-based question benchmarks (Joshi et al., 2017; Roberts et al., 2020), we introduce two evaluation protocols, *i.e,*

---

[3]Based on manual inspection of 500 examples, we found this process rarely produces incorrect examples ($\leq 1.2\%$).

| Eval Protocol | Training/Validation | Testing | Methods | Example Models | Knowledge Base |
|---|---|---|---|---|---|
| **No-KB** | {I, Q, A} | {I, Q} | End-to-end Model | PaLI, BLIP2 | - |
| **With-KB** | {I, Q, A, E} | {I, Q} | Pipeline System | CLIP→ PaLM / FiD | Wikipedia |

Table 3: **Two evaluation protocols of INFOSEEK.** The key difference is whether auxiliary data for visual entity recognition and knowledge base is available at training. I: image, Q: question, A: answer, E: queried visual entity.

*No KB* and *With KB*, to evaluate models with different information accessible from INFOSEEK. Table 3 and Figure 2 have provided a comparison for the two setups. This key design choice is made to encourage models from different families to be compared with a clear notion of what information was accessed. We note that the *No KB* protocol is more challenging than the *With KB* protocol.

**The No-KB protocol.** Models are tasked to directly predict the answer by examining the image and question, similar to traditional VQA systems. This requires the model to store world knowledge in its parameters for effective question answering. The research question focuses on how much knowledge can an end-to-end model memorize in its parameters during pre-training, and how well can it utilize this knowledge after fine-tuning? We use the standard VQA formatted data, *i.e*, {Image (I), Question(Q), Answer(A)} triplets for training / validation, and {I, Q} for testing.

**The With-KB protocol.** The goal is to analyze headroom for improvement when a viable reasoning chain is explicitly provided. Therefore, this protocol encourages an extra step of visual entity recognition, grounding the task on a knowledge base. The VQA task is transformed into a two-step pipeline, *i.e*, (1) visual entity recognition; and (2) language QA with entity information. We provide training signals to first recognize the queried visual entity and then leverage the information to query a large language model for answers, or identify relevant Wikipedia articles for extracting the answer. Specifically, we provide a 100K Wikipedia KB (articles and infobox images) that includes visual entities from INFOSEEK and top frequent entities from Wikipedia. During training and validation, *With KB* protocol provides entity labels for each queried visual entity. During testing, the model is evaluated based on the {I, Q} pairs only.

### 4.1 Models without KB Information

**Random & Prior.** Random answers sampled from the training set; The majority answer based on the question prior, which is calculated using the train-

ing set questions grouped by question 4-gram.

**PaLM (Q-only) Model.** To validate the importance of visual content in INFOSEEK, we build a question-only baseline with PaLM (540B) (Chowdhery et al., 2022), using text question as the only input and with 5-shot in-context-learning.

**BLIP2 & InstructBLIP.** We utilize two pre-trained vision-language models, *i.e*, BLIP2 (Li et al., 2023b) and InstructBLIP (Dai et al., 2023). Both models share the same architecture, which trains a Q-former Transformer that connects a frozen vision encoder (ViT-g/14) to a frozen instruction-tuned language model (Flan-T5$_{XXL}$ (Chung et al., 2022)) to output text based on an input image and text. Particularly, Instruct-BLIP fine-tunes the BLIP2 model on 26 vision-language datasets (e.g., VQAv2, OKVQA) with a text instruction prefix, and claimed to show improved zero-shot performance on unseen vision-language tasks. Following Li et al. (2023b), we fine-tune the Q-former of both models using the INFOSEEK $_{Wikidata}$, for improved performance.

**PaLI-17B & PaLI-X.** We experiment with two extra pre-trained vision-language models from the PaLI (Chen et al., 2023b,a) family given its SOTA performance. Particularly, we use PaLI-17B (ViT-e + mT5$_{XXL}$ (Xue et al., 2020)) and PaLI-X (ViT-22B (Dehghani et al., 2023) + UL2-33B (Tay et al., 2022)), which are pre-trained on WebLI (Chen et al., 2023b) with 1 billion image-text pairs. Both models, which use non instruction-tuned language models, exhibit minimal zero-shot performance on INFOSEEK. Consequently, we fine-tune both models on the INFOSEEK $_{Wikidata}$ to improve their performance.

### 4.2 Models with KB Information

In this protocol, we explicitly model the path to answer info-seeking questions with two decoupled sub-tasks: (1) recognizing the visual entity grounded to the KB and (2) textual reasoning to answer the question. A hidden benefit of such pipeline systems is improved interpretability, because it is easier to locate the source of errors by

| Model | LLM | # Params | INFOSEEK_Wikidata | | | INFOSEEK_Human | | | OK-VQA | VQAv2 |
|---|---|---|---|---|---|---|---|---|---|---|
| | | | UNSEEN QUESTION | UNSEEN ENTITY | Overall | UNSEEN QUESTION | UNSEEN ENTITY | Overall | Accuracy | Accuracy |
| Random | - | - | 0.1 | 0.1 | 0.1 | 0.2 | 0.1 | 0.1 | - | - |
| Prior | - | - | 3.9 | 2.7 | 3.2 | 0.3 | 0.3 | 0.3 | - | 32.1 [†] |
| PaLM (Q-only) | PaLM | 540B | 5.1 | 3.7 | 4.3 | 4.8 | 6.6 | 5.6 | 23.8 | 43.0 |
| BLIP2 | Flan-T5$_{XXL}$ | 12B | 14.5 | 13.3 | 13.9 | 10.0 | 8.9 | 9.4 | 54.7 | 82.3 |
| InstructBLIP | Flan-T5$_{XXL}$ | 12B | 14.3 | 13.2 | 13.7 | 10.6 | 9.3 | 9.9 | 55.5 | - |
| PaLI-17B | mT5$_{XXL}$ | 17B | 20.7 | 16.0 | 18.1 | 13.3 | 5.9 | 8.2 | 64.8 | 84.6 |
| PaLI-X | UL2$_{32B}$ | 55B | 23.5 | 20.8 | 22.1 | 12.9 | 9.3 | 10.8 | 66.1 | 86.1 |

†: Numbers adopted from Agrawal et al.

Table 4: **Results of No-KB models fine-tuned on INFOSEEK.** Baselines including Random, Prior (majority answer with 4-gram question prior), and a question-only model using PaLM (Q-only) with 5-shot prompting. VQA accuracy of models on OK-VQA (Marino et al., 2019) and VQAv2 (Goyal et al., 2017) are for comparison.

diagnosing each sub-task component.

**Sub-task #1: Visual Entity Recognition.** We follow the entity recognition task defined in OVEN (Hu et al., 2023), and use an image and a text query (e.g., "What is this building?") as model inputs, and predict entities among 100K multi-modal Wikipedia entries. Particularly, we employ the pre-trained CLIP (Radford et al., 2021) model (ViT-L/14), as our visual entity recognition model, because of its strong generalization capability. Specifically, we follow the CLIP2CLIP model described in Hu et al., to fine-tune CLIP to encode multi-modal representations (image, question) from our dataset as query, and (Wikipedia image, Wikipedia title) from the KB as candidates. We then retrieve the top $k$=5 most similar entities based on weighted cosine similarity scores computed between the query and candidates.

**Sub-task #2: Language QA with LLM or KB Reader.** Through visual entity recognition, we can now represent the queried visual information as its textual description. This enables us to investigate the language reasoning component independently to understand how much improvement a strong LLM or a KB reader can bring.

- **PaLM: Large Language Model.** We use PaLM (540B) to investigate the amount of knowledge that can be memorized in the model's parameters from pre-training on text corpora. Given a question and the queried entity name (from entity recognition), we prompt PaLM to predict the answer using 5-shot in-context examples with the prompt format: "question: This is {entity} {question} answer:".

- **Fusion-in Decoder (FiD): KB Reader.** We experiment with a SOTA retrieval-augmented

model, which reads information from a KB, to understand the value of Wikipedia articles in the KB. Specifically, the FiD (Izacard and Grave, 2020) model is employed, which takes $N$=100 retrieved articles as input and generates an answer. The model is pre-trained with a T5$_{Large}$ (Raffel et al., 2020) backbone (660M) on Natural Questions (Kwiatkowski et al., 2019) and fine-tuned on INFOSEEK. During inference, we retrieve the first 20 passages from Wikipedia for $k$=5 visual entities (from entity recognition) and feed 100 passages to FiD to generate the answer.

## 5 Experiments

### 5.1 Results for No-KB Models

**Main results.** Table 4 presents the results of end-to-end models on INFOSEEK. The best pre-trained model in this setting is PaLI-X, although the absolute number on the model's overall performance remains low. This is partially due to the fact that INFOSEEK questions often require identifying entities and retrieving specific information relevant to the question, making it a challenging task for end-to-end models. As PaLI-X is pre-trained on a large corpus with more model parameters, it demonstrates better generalization ability on the UNSEEN ENTITY split compared to PaLI-17B. Meanwhile, there remains a noticeable gap in performance on the UNSEEN QUESTION and UNSEEN ENTITY splits, indicating that models struggle with generalization to new visual entities from the training set. We also present models' results on OK-VQA (Marino et al., 2019) and VQAv2 (Goyal et al., 2017) for comparison and observe a drastic performance gap, emphasizing the difficulty of visual info-seeking questions.

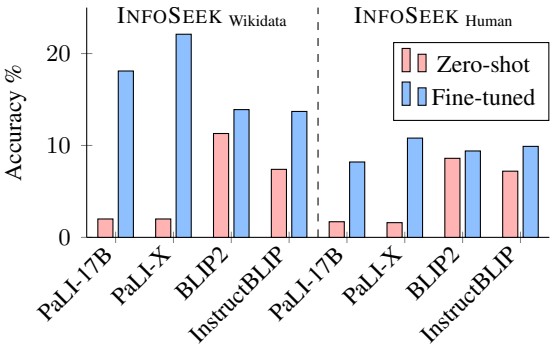

Figure 3: **Zero-shot & fine-tuned performances on** INFOSEEK. Fine-tuning on INFOSEEK elicits knowledge from PaLI models to answer fine-grained visual info-seeking questions.

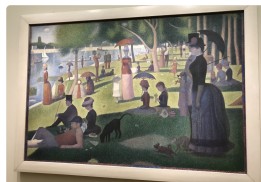 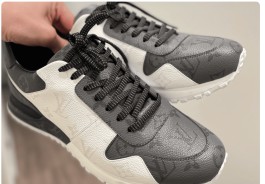

**Q**: what year was this painting created?
PaLI-17B: 1884 ✓
PaLI-X: 1884 ✓
BLIP2: 1887 ✗

**Q**: which year was this brand established?
PaLI-17B: 1915 ✗
PaLI-X: 1854 ✓
BLIP2: 1854 ✓

Figure 4: **Predictions on out-of-domain visual entities** (art & fashion) collected from real-world images by authors, using INFOSEEK fine-tuned models.

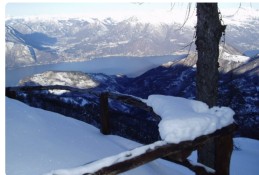 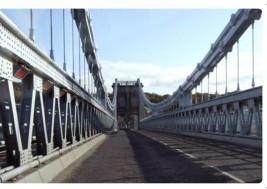

**Q**: Which body of water is this mountain located in or next to?
**A**: Lake Como
BLIP2(0-shot): lake como
InstructBLIP(0-shot): lake

**Q**: Who designed this bridge?
**A**: Thomas Telford
BLIP2(0-shot): john nash
InstructBLIP(0-shot): architect

Figure 5: **InstructBLIP(0-shot) makes less fine-grained predictions** compared to its initial model (BLIP2), after instruction-tuned on prior VQA datasets.

**Fine-tuning elicits knowledge from the model.** To demonstrate the value of INFOSEEK training data, we report the zero-shot performance of models in Figure 3. Specifically, we find that without fine-tuning, both PaLI models produce a negligible overall performance, which is significantly worse than the fine-tuned counterpart. This provides evidence to support the hypothesis that fine-tuning has helped elicit knowledge from the pre-trained PaLI models. On the other hand, BLIP2 and InstructBLIP show compelling zero-shot performance on INFOSEEK as they adopt a frozen instruction fine-tuned LLM (*i.e*, Flan-T5) and InstructBLIP is further instruction-tuned on a collection of VQA benchmarks. The performance of BLIP2 models is further improved after fine-tuning on INFOSEEK with a small number of steps, showing strong generalization results to the Human split. In Figure 10, we present examples of BLIP2 predicting the "country location" of an unseen entity (*i.e*Amberd) and show the accuracy was improved from 18% to 92% after fine-tuning, despite not seeing this entity in the training set. Finally, we conducted a real-world evaluation on out-of-domain images unavailable from the Internet (not from any models' pre-training data). Particularly, we evaluate fine-tuned PaLI with 90 questions on 30 images captured by the authors, on visual entities outside of the INFOSEEK training corpus. As a result, PaLI-17B and PaLI-X answered 22.2% and 38.9% of questions correctly. Figure 4 presents examples of PaLI and BLIP2 predictions on two out-of-domain entities (artwork and fashion product).

**Why does instruction-tuned BLIP2 obtain worse zero-shot INFOSEEK results?** One surprising finding from Figure 3 caught our attention and reveals an important criterion to be consid-

ered for future model development. We found InstructBLIP0-shot performs significantly worse than its initial checkpoint, BLIP2 (7.4 vs 11.3 on InfoSeek Wikidata), which contradicts the superior zero-shot performances of InstructBLIP in Dai et al. (2023). We conduct manual analysis and detect a common error made by InstructBLIP is its preference for generating coarse-grained predictions compared to BLIP2 (e.g., architect vs a person's name). This leads to a performance drop on INFOSEEK, which emphasizes fine-grained answers (see Figure 5). We hypothesize that this can be attributed to the instruction tuning datasets used for InstructBLIP (e.g., VQAv2 and OK-VQA), which share a less fine-grained answer distribution. Fortunately, fine-tuning on INFOSEEK Wikidata helps close the gap.

## 5.2 Results for With-KB Models

**Models with KB access perform better.** Table 5 presents the results for pipeline models with access to knowledge base (KB) information, along with the best results from the No-KB setting for reference. Notably, the pipeline models outperform the best No-KB models on the challenging INFOSEEK Human split significantly. This highlights the pipeline systems' ability to answer visual info-

| Model | INFOSEEK Wikidata | INFOSEEK Human | ENTITY Accuracy |
|---|---|---|---|
| Best No-KB | 22.1 | 10.8 | - |
| **With-KB Setting** | | | |
| CLIP → PaLM | 20.1 | 15.2 | 22.2 |
| CLIP → FID | 19.3 | 18.2 | |
| Oracle → FID | 52.0 | 45.6 | 100 |

Table 5: **Results of With-KB setting.** CLIP → PaLM/FID: a two-stage pipeline system (visual entity recognition → text QA). PaLM: 5-shot prompting. FID: Fusion-in Decoder to read from KB using T5$_{large}$. Oracle: an artificial upper-bound using oracle entities.

| Model | TIME (Acc.) | NUMERICAL (Relaxed Acc.) | STRING (Acc.) |
|---|---|---|---|
| **No-KB Setting** | | | |
| Prior | 0 | 4.4 | 5.0 |
| PaLM (Q-only) | 0 | 11.4 | 4.0 |
| InstructBLIP | 7.9 | 7.5 | 17.8 |
| BLIP2 | 6.9 | 5.8 | 18.5 |
| PaLI-17B | 3.8 | 18.4 | 27.4 |
| PaLI-X | 7.7 | 16.1 | 30.0 |
| **With-KB Setting** | | | |
| CLIP → PaLM | 12.5 | 27.7 | 21.7 |
| CLIP → FiD | 12.3 | 23.4 | 23.9 |

Table 6: **Results w.r.t. each question types** on the INFOSEEK$_{Wikidata}$ val set of unseen question split, showing a big headroom for improvements on TIME and NUMERICAL for all end-to-end models.

seeking questions by effectively utilizing visual recognition and language reasoning, specifically using the names of visual entities to convey information across modalities. When comparing the two Language QA models, we observe that the FiD model, which reads the Wikipedia article, achieves the highest generalization performance on INFOSEEK $_{Human}$ by a significant margin. This suggests that access to relevant text content plays a crucial role in answering visual info-seeking questions.

**Large headroom for improvement.** Table 5 demonstrates an artificial upper-bound (Oracle → FiD) on INFOSEEK, indicating substantial room for performance improvement if an oracle entity recognition model were available. By simulating the visual entity recognition's accuracy improvement (from 22% using CLIP to 100%), the INFOSEEK accuracy can be improved from ∼20% to ∼50%, within the same FiD model.

**Analysis on each question type.** Table 6 shows a breakdown of results under different question types, evaluated on INFOSEEK $_{Wikidata}$. Comparing **No KB** and **With KB** models, we found that end-

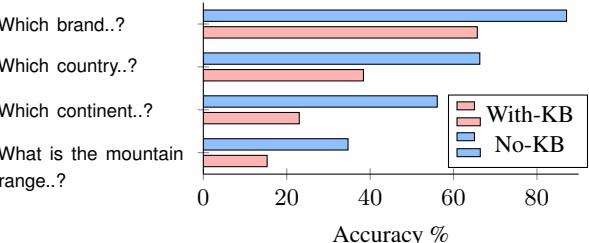

Figure 6: No-KB (PaLI-17B) outperforms With-KB (CLIP→FiD) models on questions that query less fine-grained attributes.

to-end models such as PaLI, have a short barrel on fine-grained knowledge-intensive questions (*i.e*, TIME and NUMERICAL). It can perform well on other questions, which are more about querying attributes or resolving relations between entities (see Figure 6). Comparing **With KB** models, PaLM and FiD perform on par with each other on this automated evaluation data. However, when evaluated on the natural info-seeking human queries, FiD has a better generalization, outperforming PaLM on TIME (21.5 vs 14.6) and NUMERICAL (25.6 vs 21.3) questions from INFOSEEK $_{Human}$ significantly. One possible reason is that natural info-seeking questions written by people focus more on very fine-grained information, which is rare and hard to memorize for PaLM. In contrast, FiD can leverage Wikipedia articles to predict answers. Finally, we analyze the performance of different models according to the visual entity popularity and found unique advantages of end-to-end models (see Appendix).

**Performance on Head vs. Tail entities.** Although pipeline models with KB access are overall stronger, surprisingly, we observe that end-to-end models have a unique advantage for info-seeking VQA, particularly on the tail entities. Figure 7 presents a comparison of models, with group-wise performances on Wikipedia entities that are least popular (less monthly page views) to most popular (more monthly page views). The histogram is generated based on the average monthly Wikipedia pageviews in 2022, following (Mallen et al., 2022). Surprisingly, the results show that PaLI-17B outperforms the pipeline systems by a large margin on the tail entities, particularly for questions related to geographical information. We show some qualitative examples in Figure 9, for entities from baskets of different monthly page views. This suggests that there are many different routes to answer

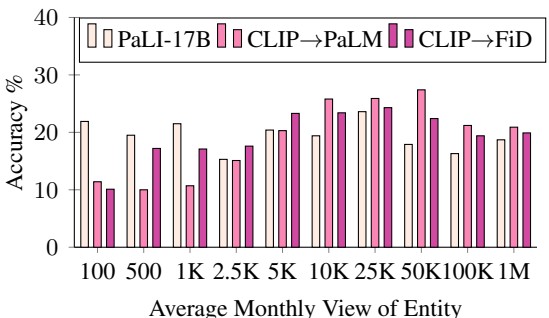

Figure 7: **INFOSEEK results w.r.t. visual entities of different popularity.** End-to-end model outperforms pipeline systems on tail entities (low monthly pageviews) but overturned on more popular entities (high monthly pageviews).

visual info-seeking questions and that pipeline systems that rely on an explicit decomposition of the VQA task may be redundant and susceptible to error propagation from the entity linking stage. Whereas for end-to-end models such as PaLI, it is flexible to decide which route of reasoning is more appropriate to answer a given question. For example, one can answer geographical questions without knowing the identity of the visual entity, if other relevant visual clues are presented. Meanwhile, on the more popular head visual entities, a clear trend emerged showing that pipeline systems outperform end-to-end PaLI by a big margin.

## 6 Related Work

**Pre-trained Vision Language Models.** There has been significant growth in the development of vision-language models pre-trained on large-scale image-text datasets (Lu et al., 2022; Bao et al., 2021; Wang et al., 2022; Zhou et al., 2020; Radford et al., 2021). One line of research aims to augment a pre-trained language model with visual modality by learning a mapping from an external visual encoder to the frozen large language model (Alayrac et al., 2022; Li et al., 2023b; Koh et al., 2023), to fully leverage textual knowledge from the language model (Xu et al., 2023; Dai et al., 2023; Liu et al., 2023; Zhu et al., 2023; Ye et al., 2023).

**Knowledge-based VQA Models.** Various approaches have been proposed to address knowledge-based VQA tasks (Marino et al., 2019) by incorporating external knowledge into vision-language models. One approach is to retrieve information from an external KB (Marino et al., 2021; Hu et al., 2022b; Wu and Mooney, 2022) and em-

ploy a model (Izacard and Grave, 2020) to perform language QA (Gui et al., 2022; Lin et al., 2022). Other approaches transform the image into a text caption and use an LLM (Brown et al., 2020; Chowdhery et al., 2022) to answer questions (Yang et al., 2022; Hu et al., 2022a). We utilize both approaches to study the ceiling for improvement on INFOSEEK with the OVEN model (Hu et al., 2023).

Another concurrent work (Mensink et al., 2023) investigates similar challenges but emphasizes scalability and relies on model-generated annotations, as opposed to our human-annotated info-seeking queries.

## 7 Conclusion

We introduced INFOSEEK, a large-scale VQA dataset that focuses on answering visual information seeking questions. With INFOSEEK, we found that current state-of-the-art pre-trained visual-language models struggle to answer visual info-seeking questions requiring fine-grained knowledge, such as questions about time and numerical information of a visual entity. Our analysis using pipeline systems, which ground visual entities to an external knowledge base, suggests that incorporating fine-grained knowledge into the pre-training process holds significant potential to improve end-to-end pre-training models.

## 8 Limitation

INFOSEEK is limited to English language and future research could expand it to a multilingual setting, leveraging articles in Wikipedia supported in other languages. While the primary focus of this work is on knowledge derived from Wikipedia, future investigations could explore extensions to other domains, such as medical information, and artwork, and incorporate emerging updates in Wikipedia (Iv et al., 2022).

## Acknowledgement

We thank Jialin Wu, Luowei Zhou for reviewing an early version of this paper. We thank Xi Chen for providing different variants of PaLI pre-trained checkpoints. We also thank Radu Soricut, Anelia Angelova, Fei Sha, Andre Araujo, Vittorio Ferrari, Wei Xu, Kartik Goyal for valuable discussions and feedback on the project. Yang Chen is partially funded by the NSF (IIS-2052498).

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

| | #UNSEEN QUESTION/ENTITY | #Total | Question Type TIME/NUM./STR. | #Entity |
|---|---|---|---|---|
| Train | - / - | 934,048 | 4.4/20.4/ 75.2% | 5,549 |
| Val | 18,656/54,964 | 73,620 | 4.6/ 21.6/ 73.8% | 1,794 |
| Test | 98,901/249,079 | 347,980 | 4.8/22.9/72.3% | 8,905 |
| Human | 3,248/5,683 | 8,931 | 26.8/ 26.4/46.8% | 806 |

Table 7: **INFOSEEK Dataset statistics.** Average question per image rate is 1.4 and 1.0 for Wikidata and Human split, respectively.

## A  Details of the Dataset.

In this section, we provide more details of the human annotation quality control and automatic data generation process. We summarize the statistics of INFOSEEK in Table 7 and show question prefix distribution in Figure 11 and entity distribution in Figure 12.

### A.1  Human Annotation Quality Control

**Instruction and Training.** We hire 30 full-time in-house annotators to collect questions and answers in INFOSEEK_Human. Annotators are native English speakers in the U.S. and are aware of the purpose of the collected data. To ensure the quality of annotations in the INFOSEEK_Human dataset, a comprehensive training process was designed and implemented for our annotators. This process involved a pilot study, in which annotators read the instructions and annotated a few sample examples, followed by a tutorial session and a quiz. The tutorial was conducted through an online video session and provided a comprehensive overview of the instructions while addressing common mistakes identified in the pilot study. Only annotators who passed the quiz were selected to work on the main task, with 30 annotators completing the training. We hire annotators at $17.8 per hour, which is higher than the minimum wage in the U.S., to fairly compensate annotators for their time and effort. The average completion time for stages one and two of the annotation task was 12 and 10 minutes, respectively. A screenshot of the annotation interface is provided in Figure 13.

### Annotation Procedure

*Stage 1 (Question Writing)*: As shown in Figure 13 (Top), annotators are shown with images of a visual entity on the left-hand side with a short description of the entity from Wikipedia below. On the right, we show a list of Wikipedia section titles of the entity and ask annotators to write relevant questions next to the section title. We prevent annotators from asking binary questions,

| | Correct | Incorrect |
|---|---|---|
| Percentage | 95% | 5% |

Table 8: Expert judgments of answer accuracy based on a sample of 200 examples from INFOSEEK _Human.

asking visual attributes (such as color), writing questions by rephrasing the description, copying entity names and section titles into the question, and avoiding writing ambiguous questions.

*Stage 2 (Answer Labeling)*: As shown at the bottom of Figure 13, annotators are present with info-seeking questions to the entity collected from Stage 1 and a Wikipedia link of the entity. For each question, annotators are asked to find a short span of answers (less than 10 words) from the Wikipedia page. They are asked to answer two questions: (1) "Can you derive the answer from the given Wikipedia page?" and (2) "What is the type of this question?" and select from three options (TIME, NUMERICAL, OTHERS). For each answer, they will then fill in the answer box (TIME: `[year, month, day]`, NUMERICAL: `[min, max, unit]`, OTHERS: `[string]`) and copy paste a short sentence from Wikipedia that contains the answer to the evidence section. We decided to exclude questions without answer spans from Wikipedia following TyDiQA-GoldP as the dataset is already hard enough and reserve these questions for future work.

**Expert Feedback and Correction.** Expert annotators provided regular feedback during annotation and conducted thorough post-annotation verification. The data was split into three batches, with annotators flagged and provided feedback for those who consistently made similar mistakes. After the completion of stage 1, questions that revealed the entity name, asked about the color or shape of an object, or were binary were automatically rejected. After stage 2, three expert annotators reviewed and processed the question-answer pairs, removing unqualified pairs and verifying the answer span from the annotated evidence sentence. Rejected pairs may have included questions that were not answered by the annotated answer or were too general and resulted in an ambiguous answer. The expert annotators also corrected the question type annotation and edited the answer span into the correct format, such as adding units for numerical questions or shortening long answer spans that exceeded ten tokens. Finally, the expert annota-

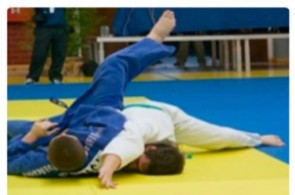 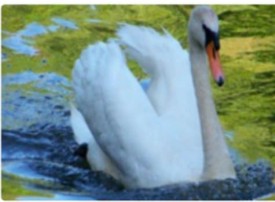 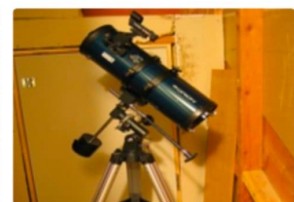 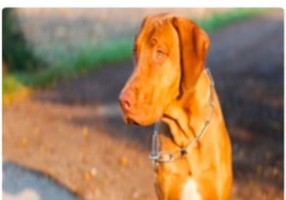

- **Q**: Who is the founder of this sport?
- **A**: Kanō Jigorō

- **Q**: What is the length of this bird in centimetre?
- **A**: 120-170

- **Q**: Who is the inventor of this object?
- **A**: James Gregory

- **Q**: What is the country of origin of this animal?
- **A**: Rhodesia

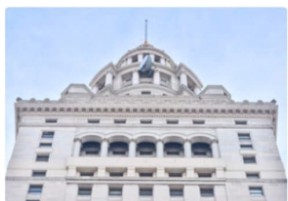 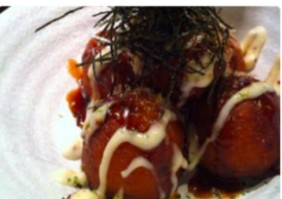 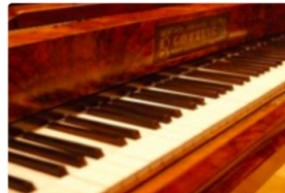 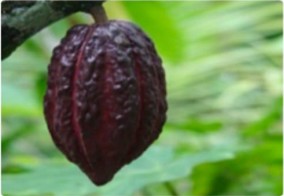

- **Q**: In which year did this building officially open?
- **A**: 1930

- **Q**: Which year was this food invented?
- **A**: 1935

- **Q**: What is the highest note this item can play?
- **A**: C8

- **Q**: Where is this plant native to?
- **A**: Ecuador

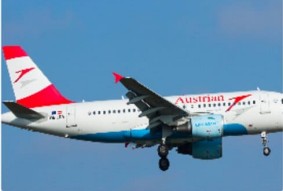 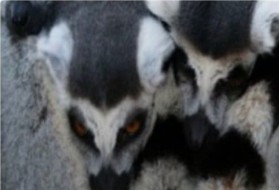 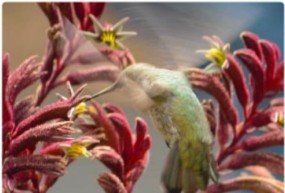 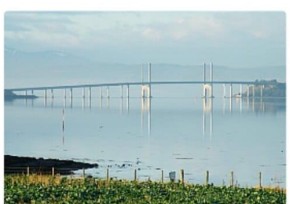

- **Q**: What is the cruise speed of this aircraft (kilometer per hour)?
- **A**: 828

- **Q**: Which country is this animal endemic to?
- **A**: Madagascar

- **Q**: How heavy does this bird typically grow up to in terms of gram?
- **A**: 4.3

- **Q**: Who is the structure engineer of this bridge?
- **A**: Hellmut Homberg

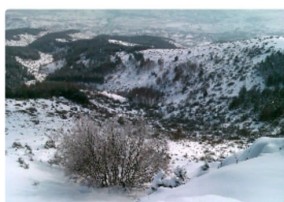 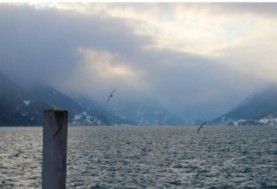 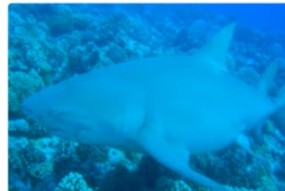 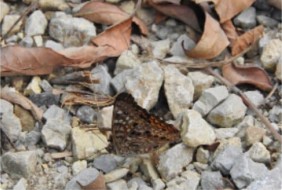

- **Q**: What country does this mountain belong to?
- **A**: North Macedonia

- **Q**: What is the area in square kilometers of this lake?
- **A**: 29.8

- **Q**: What is this fish named after?
- **A**: lemon

- **Q**: What is the closest parent taxonomy of this insect?
- **A**: Asterocampa

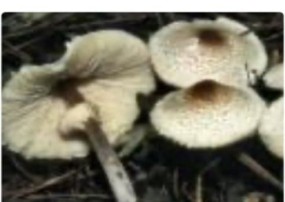 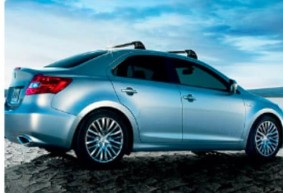 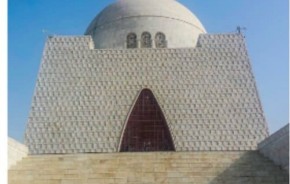 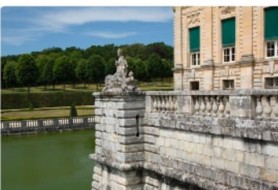

- **Q**: What is the basionym of this plant?
- **A**: Agaricus sect. Lepiota

- **Q**: What is the manufacturer of this vehicle?
- **A**: Suzuki

- **Q**: Who does this place commemorate?
- **A**: Mohammad Ali Jinnah

- **Q**: In which year was this building built?
- **A**: 1661

Figure 8: Random examples from the training set of INFOSEEK Wikidata.

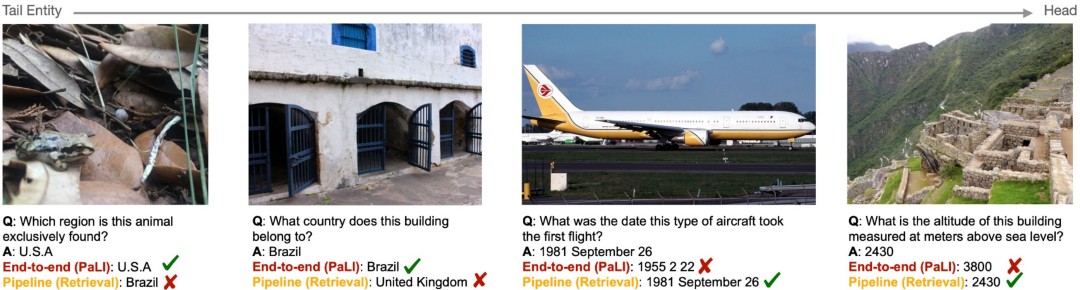

Figure 9: **Examples** of predictions of PaLI-17B and CLIP→FID on INFOSEEK (left to right shows tail to head entities).

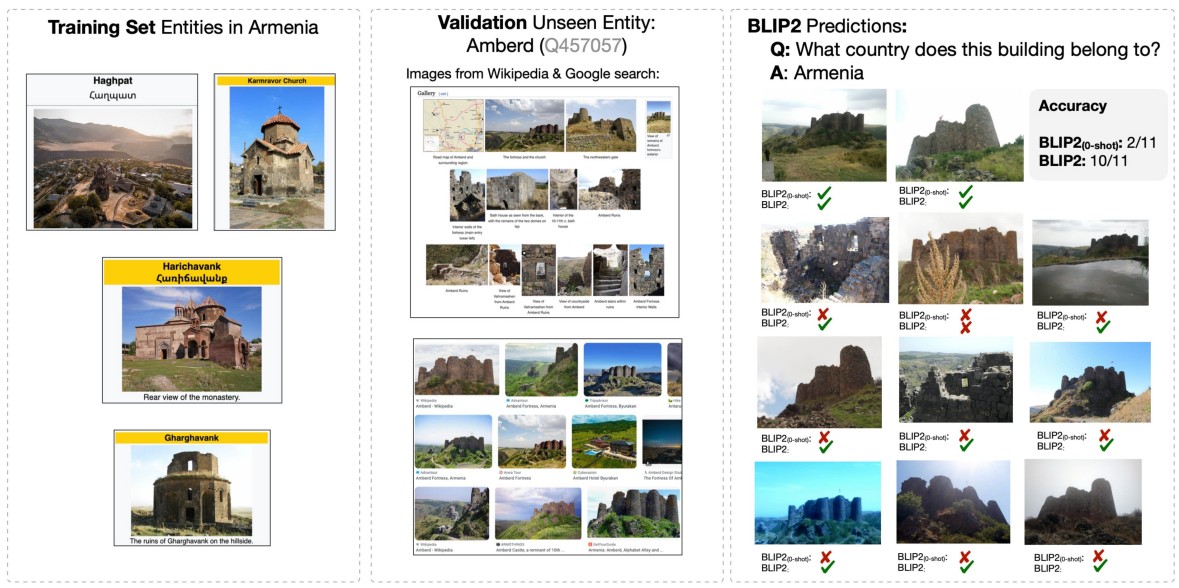

Figure 10: **Examples** of predictions of BLIP2$_{(0\text{-}shot)}$ and BLIP2 on INFOSEEK (Entity: Q457057). Fine-tuning improves the accuracy from 2/11 to 10/11, despite it being an UNSEEN ENTITY (not in the training set). We show training set entities that are located in Armenia and images of Amberd on the internet.

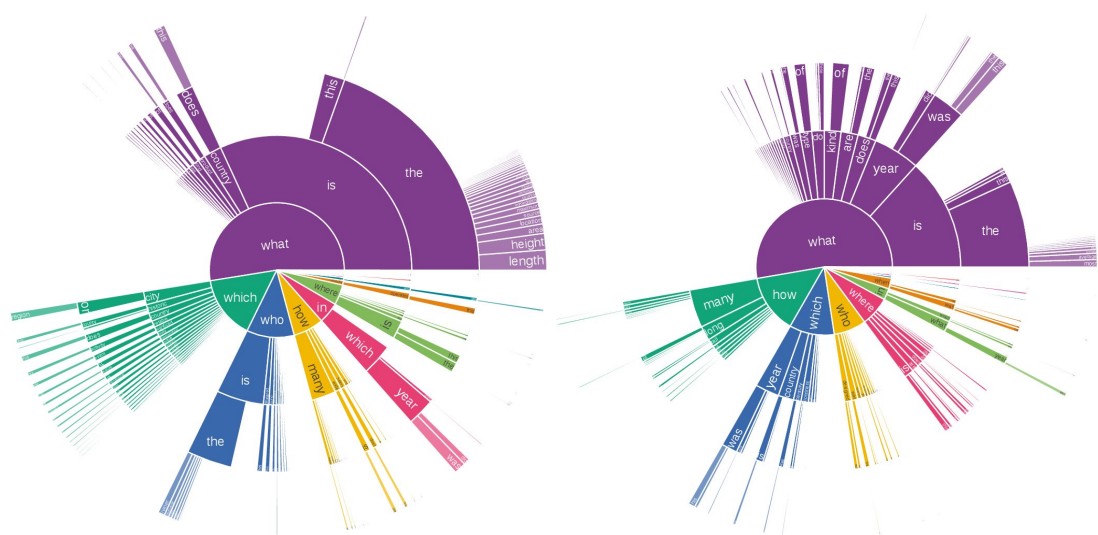

Figure 11: Question prefix distribution in INFOSEEK Wikidata (left) and INFOSEEK Human (right).

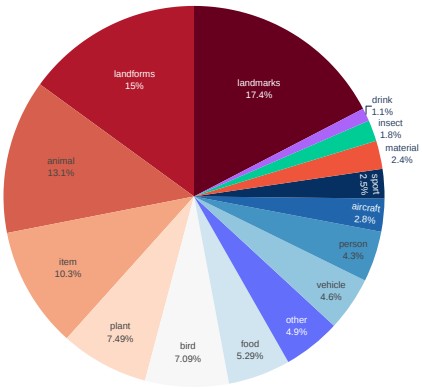

Figure 12: Distribution of the entities in INFOSEEK (Grouped by their super category).

tors reviewed the image-question-answer triples to reject bad images or clarify the question when multiple objects were present in the image. For example, a building was specified when multiple buildings were present in the image. On average, it took 1.5 hours to verify 1000 triples, as the majority of images contained a single object.

Following TyDiQA (Clark et al., 2020), we analyze the degree to which the annotations are correct instead of the inter-annotator agreement since the question may have multiple correct answers. In Table 8, human experts carefully judged a sample of 200 examples from INFOSEEK _Human_ split. For each example, the expert reads through the Wikipedia page of the queried visual entity and finds the answer to the question. They then indicate whether the annotated answer is correct. We take the high accuracy (95%) as evidence that the quality of the dataset offers a valuable and reliable signal for evaluating visual info-seeking models.

## A.2 Filtering and Subsampling

**Filtering.** To test the models' ability to answer visual information-seeking questions that require fine-grained knowledge, which can be learned from the pre-training corpus such as Wikipedia, we need to verify the consistency of answers between Wikidata and Wikipedia. Given that Wikidata and Wikipedia are crowd-sourced independently, some QA pairs created from Wikidata may not be present in the Wikipedia article or may have different answers. Therefore, we filtered out QA pairs where the answer could not be found in the Wikipedia article of the entity. We performed an exact string match to verify answers for string questions and

used fuzzy string matching [4] with a substring ratio greater than 0.9 if an exact match could not be found. For time questions, we applied an exact match to verify the year, month, and date. In some cases, the year of construction of a building varied by a year, so we allowed a +/- 1 year deviation for the time question. For numerical questions, we used exact matching to verify the numbers in the article. However, in many cases, the units were different (meters or inches), or a range with a minimum and maximum was given. We used regular expressions to extract the number or range from the Wikipedia article and filter out the QA pairs if it is counted as incorrect based on the "Relaxed accuracy" in Section 3.3 in the main text. Based on a manual analysis of 200 randomly sampled QA pairs, we found that 97% of the answers of the IN-FOSEEK _Wikidata_ could be found in the Wikipedia article.

**Subsampling Questions.** In order to achieve a more diverse set of questions in INFOSEEK, we applied a subsampling method to address the skewed distribution of crowd-sourced knowledge triples in Wikidata. The method followed the approach used in Zhong et al. (2022). This involved defining $P(r, c)$ as the percentage of triples that contain the relation $r$ and the subject entity's category as $c$. The $P'(r, c) = 1/|(r, c)|$ was calculated as the average probability of a relation-category pair and Image-Q-A triples were removed with increasing likelihood based on the probability $r = 1 - min(1, P(r, c)'/P(r, c))^{1/2}$. Additionally, the same subsampling method was applied to balance the answer distribution for each relation. This resulted in the question prior baseline achieving a relatively low score (3.2) in INFOSEEK _Wikidata_, as shown in Table 4 in the main text.

## A.3 Evaluation Metric.

There are three types of questions, *i.e*, STRING, TIME, and NUMERICAL, which are evaluated differently. Particularly, we adopt the VQA accuracy (Goyal et al., 2017; Marino et al., 2019) against multiple references for STRING and TIME questions, and utilize *Relaxed Accuracy* (Methani et al., 2020; Masry et al., 2022) for NUMERICAL questions. For STRING questions, we use the alias of answers from Wikidata as multiple references for INFOSEEK _Wikidata_ (#avg = 4.5), and the human-annotated multiple

---

[4] SequenceMatcher from `difflib` library

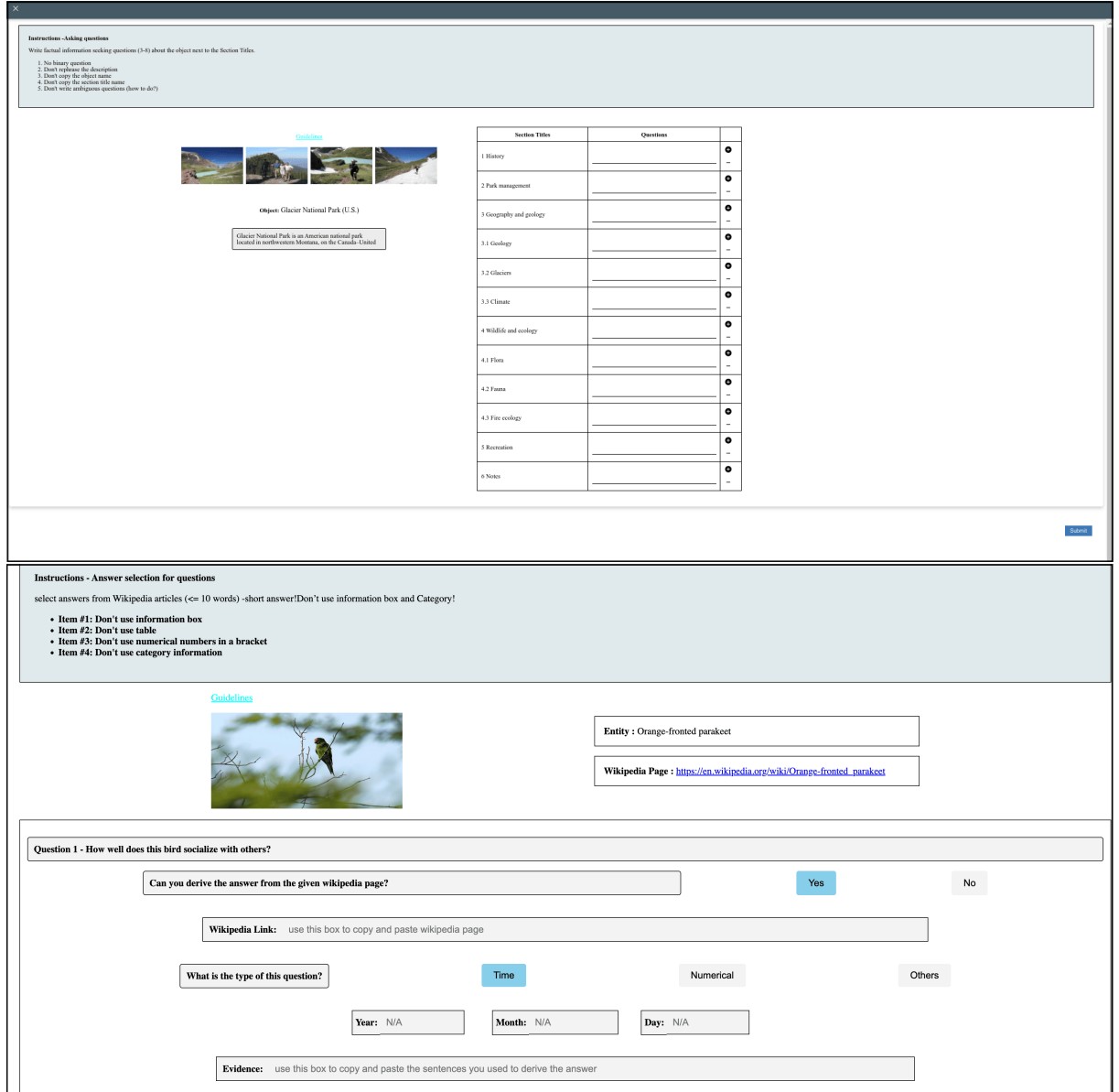

Figure 13: Annotation Interface for Stage 1 (Top) and Stage 2 (Bottom).

references for INFOSEEK Human (#avg = 2.4).

Exact Match: Correct if the prediction matches any one of the references exactly.

- prediction="USA", references=["USA", "U.S.", "United States of America", ...] → ✓

Exact Match: Correct if the prediction matches any one of the references exactly.

- prediction="1991", references=["1990", "1991", "1992"]) → ✓
- prediction="1991 6 11", references=["1991 6 11", "1991 June 11", "11 June 1991", ...] → ✓

For TIME questions, the answer references account for different date formats of year/month/day. Meanwhile, we perform a relaxed match (with a one-year error tolerance) to measure the model's prediction, because it is quite often that historical events are only associated with estimated time.

For NUMERICAL questions, the exact match would not be able to handle the case where a range (e.g., a pair of minimum and maximum values) is provided as annotated ground truth. To account for this, we make a slight modification to the Relaxed Accuracy with a 10% tolerance range.

Relaxed Accuracy: correct if the prediction is in the reference range or the prediction range overlaps with the reference range of more than 50%
1) ref_min ≤ pred ≤ ref_max;
2) IoU([pred_min, pred_max], [ref_min, ref_max]) ≥ 50%

• reference= 10 cm → reference= [9, 11] # 10% tolerance
• prediction= 10, reference= [9, 11] → ✓
• prediction= [5, 6], reference= [9, 11] → ×

A single value prediction is counted as correct if it falls within the answer range, and a range prediction is correct if the intersection-of-union between the prediction and answer is greater than or equal to 50%. Finally, we calculate the accuracy for each data split (UNSEEN QUESTION and UNSEEN ENTITY), and take the harmonic mean of them as the overall accuracy.

### A.4 Image Sources.

Image Recognition (or Retrieval) Datasets: ImageNet21k-P (Russakovsky et al., 2015; Ridnik et al., 2021), iNaturalist2017 (Van Horn et al., 2018), Cars196 (Krause et al., 2013), SUN397 (Xiao et al., 2010), Food101 (Bossard et al., 2014), Sports100 (Gerry, 2021), Aircraft (Maji et al., 2013), Oxford Flower (Nilsback and Zisserman, 2008), Google Landmarks v2 (Weyand et al., 2020).

## B Implementation details of the baseline systems

In this section, we provide complete implementation details of baseline models for the INFOS-EEK task. We summarize hyperparameters for fine-tuning in Table 9.

### B.1 without-KB Models

**PaLI and PaLI-X.** We fine-tuned a 17B PaLI (Chen et al., 2023b) and 55B PaLI-X (Chen et al., 2023a) on INFOSEEK training set using the "answer in en: [question] <extra_id_0>" prompt.

**BLIP2 and InstructBLIP.** We fine-tuned a BLIP2 (Li et al., 2023b) and InstructBLIP (Dai et al., 2023) on INFOSEEK training set using the "Question: [question] Short answer:" prompt with the LAVIS library (Li et al., 2023a). The length penalty is set to -1. Since BLIP2 models present zero-shot capabilities on INFOSEEK, we employ early stopping to prevent over-fitting on the training set based on the performance on the validation set.

**OFA.** We fine-tuned the OFA$_{large}$ (Lu et al., 2022) model for 20k steps. During inference, we apply beam search decoding with a beam size set to 5. OFA achieves 11.7 and 4.0 on INFOSEEK Wikidata and Human split, respectively.

**mPLUG-owl.** We fine-tuned mPLUG-owl (Ye et al., 2023) for 10k steps with a learning rate of 2e-4 and batch size of 1 using LoRA (Hu et al., 2021). mPLUG-owl achieves 7.7 on INFOSEEK Human split.

**PaLM(Q-Only).** We use PaLM 540B (Chowdhery et al., 2022) in-context learning under the 5-shot setting with the following prompt:

```
Please answer the following question.
question: {Question_1}. answer: { Answer_1}.
...
question: {Question_i}. answer:
```

### B.2 With-KB Models

| | PaLI | PaLI-X | (Instruct)BLIP2 | OFA | FID |
|---|---|---|---|---|---|
| Optimizer | Adafactor | Adafactor | Adam | Adam | Adafactor |
| Batch size | 128 | 128 | 16 | 512 | 64 |
| Train steps | 10k | 800 | 400 | 20k | 200 |
| LR | 1e-4 | 1e-4 | 5e-5 | 5e-5 | 2e-4 |
| LR scheduler | linear decay | constant | constant | polynomial decay | constant |
| Warmup steps | 1000 | 1000 | - | 1000 | - |
| Image size | 224 | 224 | 224 | 480 | - |
| Beam size | 5 | 5 | 5 | 5 | 5 |
| Vision backbone | ViT-e | ViT-22B | ViT-g | ResNet152 | - |
| LM backbone | mT5$_{XXL}$ | UL2-32B | Flan-T5$_{XXL}$ | BART$_{large}$ | T5$_{large}$ |
| #Params | 17B | 55B | 12.1B | 0.4B | 0.4B |
| Computing | 32 TPU$_{v4}$ | 64 TPU$_{v4}$ | A40 | 8 A100 | 64 TPU$_{v4}$ |
| Time | 6 hours | 1 hour | 1 hour | 48 hours | 1 hour |

Table 9: Hyperparameters for fine-tuning models on INFOSEEK.

**PaLM.** We use PaLM 540B (Chowdhery et al., 2022) in-context learning under the 5-shot setting with the prompt present below. The Entity_1 is the gold entity provided in the training set (with KB setting). The Entity_i is the top-1 prediction from the entity linking stage of the queried image.

```
Please answer the following question.
question: {This is Entity_1. Question_1}. answer: { Answer_1}.
...
question: {This is Entity_i. Question_i}. answer: {
```

**FID.** The T5$_{large}$ FID (Izacard and Grave, 2020) model was fine-tuned in two stages using 100 passages with a maximum input length of 192 tokens. To form synthetic training data with (passage, question, answer) triples, we combine oracle entity passage with linked entity (from EntLinker) passages. We fine-tune the model on Natural Questions (Kwiatkowski et al., 2019) for 10k steps and then continue to fine-tune it on INFOSEEK for 200 steps with a batch size of 64.

```
question: This is Entity. Question. context: Passage
```

# C   Additional Experiment Results

**Complete numbers for With-KB Models.** We show the complete results for With-KB models in Table 10.

**Complete numbers for INFOSEEK Wikidata Validation set.** We show the complete Validation results for Without-KB and With-KB models in Table 11 and question type score of unseen entity split in Table 12.

## C.1   OK-VQA Annotation Guidelines

Five adult annotators each annotate 100 examples (500 in total) sampled from the OK-VQA training set. Annotators are instructed to categorize each example into one of three categories (see Table 13).

| Model | # Params | Components use KB | INFOSEEK$_{Wikidata}$ | | | INFOSEEK$_{Human}$ | | |
|-------|----------|-------------------|-----------------|---------------|---------|-----------------|---------------|---------|
| | | | UNSEEN QUESTION | UNSEEN ENTITY | Overall | UNSEEN QUESTION | UNSEEN ENTITY | Overall |
| CLIP → PaLM | 540B | CLIP | 21.9 | 18.6 | 20.1 | 15.6 | 14.9 | 15.2 |
| CLIP → FiD | 1B | CLIP & FiD | 20.7 | 18.1 | 19.3 | 18.9 | 17.6 | 18.2 |

Table 10: INFOSEEK full results on **With-KB** setting.

| Model | INFOSEEK$_{Wikidata}$ | | |
|-------|-----------------|---------------|---------|
| | UNSEEN QUESTION | UNSEEN ENTITY | Overall |
| **Without-KB Setting** | | | |
| Prior | 4.6 | 2.5 | 3.2 |
| PaLM (Q-only) | 5.5 | 4.2 | 4.8 |
| InstructBLIP | 15.0 | 14.0 | 14.5 |
| BLIP2 | 15.0 | 14.2 | 14.6 |
| PaLI-17B | 24.2 | 16.7 | 19.7 |
| PaLI-X | 25.8 | 22.4 | 24.0 |
| **With-KB Setting** | | | |
| CLIP → PaLM | 22.7 | 18.5 | 20.4 |
| CLIP → FiD | 23.3 | 19.1 | 20.9 |
| Oracle → FiD | 52.1 | 53.0 | 52.5 |

Table 11: INFOSEEK full results on Wikidata **validation** set.

| Model | TIME (Acc.) | NUMERICAL (Relaxed Acc.) | STRING (Acc.) |
|-------|-------------|--------------------------|---------------|
| **No-KB Setting** | | | |
| Prior | 0 | 3.5 | 2.3 |
| PaLM (Q-only) | 4.6 | 11.0 | 2.7 |
| InstructBLIP | 6.6 | 8.2 | 16.1 |
| BLIP2 | 5.6 | 6.0 | 17.0 |
| PaLI-17B | 1.0 | 14.8 | 18.2 |
| PaLI-X | 8.1 | 17.2 | 24.8 |
| **With-KB Setting** | | | |
| CLIP → PaLM | 17.8 | 21.3 | 17.7 |
| CLIP → FiD | 13.8 | 15.2 | 20.5 |

Table 12: **Results w.r.t. each question types** on the INFOSEEK$_{Wikidata}$ val set of unseen entity split.

| Question Category | Percentage |
|-------------------|------------|
| Answered directly by looking at the corresponding image | 50.8% |
| Answered without looking at the image (Q-only) | 20% |
| Requiring a Google search for an answer | 29.2% |

Table 13: OK-VQA annotation results.