# OpenReview forum: "Can Pre-trained Vision and Language Models Answer Visual Information-Seeking Questions?"
_EMNLP/2023/Conference — EMNLP 2023 Main_

### Official Review · Reviewer_Vbdw · 2023-08-01

**Soundness:** 4

**Excitement:**

3: Ambivalent: It has merits (e.g., it reports state-of-the-art results, the idea is nice), but there are key weaknesses (e.g., it describes incremental work), and it can significantly benefit from another round of revision. However, I won't object to accepting it if my co-reviewers champion it.

**Paper Topic And Main Contributions:**

This paper presents a VQA dataset INFOSEEK, that focuses on visual info-seeking questions. Besides, this paper analyze the ability of state
of-the-art models to answer visual info-seeking questions.

**Questions For The Authors:**

1. Your dataset is open-ended or multiple-choice? why the random model can achieve 0.1 or 0.2 accuracy?
2. Can you provide some methods to fix information-seeking questions (as for the sota model just achieve almost 20% accuracy)?

**Reasons To Accept:**

1. This paper presents a new vqa dataset.
2. The experiments are abundant.

**Reasons To Reject:**

Maybe you can provide a method to improve the performance on your dataset.

**Reproducibility:**

2: Would be hard pressed to reproduce the results. The contribution depends on data that are simply not available outside the author's institution or consortium; not enough details are provided.

**Reviewer Confidence:**

2: Willing to defend my evaluation, but it is fairly likely that I missed some details, didn't understand some central points, or can't be sure about the novelty of the work.

---

> ### Author Rebuttal · Authors · 2023-08-28
>
> **(Clarification) Q1 : "Your dataset is open-ended or multiple-choice? why the random model can achieve 0.1 or 0.2 accuracy?"**
>
> Our dataset consists of open-ended questions that require short text answer spans. Further illustrations and examples can be found in Figure 7 of the manuscript. As for the random baseline achieving 0.1 to 0.2 accuracy, it is a result of sampling answers from the training set. Due to the nature of the data, some answers happen to be correct purely by chance (e.g., common country name).
>
> **Q2: "Can you provide some methods to fix information-seeking questions (as for the sota model just achieve almost 20% accuracy)?"**
>
> - **InfoSeek**: The primary focus of our paper is to identify and quantify the performance gap in current SOTA models on visual information-seeking questions. Our goal is to design the task and introduce a dataset for future research and outlining the limitations of existing models.
> - **End-to-end & Retrieval Model**: We believe that emerging pre-trained multimodal models equipped with more efficient pre-training computation (without-KB setting) or retrieval-augmented methods (with-KB setting) will show significant improvements in accuracy.
> - **Retrieval methods we tried**: To offer additional context on the room for improvement, we have estimated an upper bound of over 50% accuracy with a retrieval-based models (Table 5). This suggests an improved CLIP-based image retrieval model could lead to substantial improvement on InfoSeek.

---

### Official Review · Reviewer_fHav · 2023-08-02

**Soundness:** 3

**Excitement:**

3: Ambivalent: It has merits (e.g., it reports state-of-the-art results, the idea is nice), but there are key weaknesses (e.g., it describes incremental work), and it can significantly benefit from another round of revision. However, I won't object to accepting it if my co-reviewers champion it.

**Paper Topic And Main Contributions:**

A visual question answering dataset is introduced in this paper for visual information-seeking questions. Various pre-trained multi-modal models are evaluated on this dataset and fine-tuned by using fine-grained knowledge, validating the challenge of the dataset in posing intricate questions.

**Questions For The Authors:**

1) It seems odd that 70.8% OK-VQA questions can be answered by adults without search engines. How did you come to this conclusion? What were your experimental configurations? What kind of questions and images were involved? All these details are missing;

2) During question writing, how many images are collected? How many questions are written for one image? It is claimed that 3-5 questions for a visual entity, yet an image involves multiple visual entities and how many questions are generated for one image is not clarified. Additionally, the entity-related Wiki articles have not been shown to the annotators during question generation, how to promise the answers to their questions can be extracted from these articles?

3) INFOSEEK_Wikidata is claimed to be automatically generated, while its template generation and refining are both implemented by humans. Additionally, I’m not sure whether a dataset can be claimed as automatically generated if no language models are involved in generation.

4) What are the potions of unseen entities and unseen questions in two data splits? A dataset is naturally challenging if part of its information is masked. It’s unclear the challenge of INFOSEEK_Wikidata comes from its intricate questions or its unseen information.

**Reasons To Accept:**

1) The paper is mostly well organized and easy to follow;

2) The novel dataset may illuminate the VQA research with cross-modal knowledge;

3) The experimental evaluations are relatively satisfied.

**Reasons To Reject:**

1) Most efforts in building the new dataset are contributed by humans, which is consuming, nonreproducible, and potentially nonobjective;

2) Some descriptions seem obscure or not rigorous;

3) The soundness of the new dataset needs a clarification.

**Reproducibility:**

3: Could reproduce the results with some difficulty. The settings of parameters are underspecified or subjectively determined; the training/evaluation data are not widely available.

**Reviewer Confidence:**

3: Pretty sure, but there's a chance I missed something. Although I have a good feel for this area in general, I did not carefully check the paper's details, e.g., the math, experimental design, or novelty.

---

> ### Author Rebuttal · Authors · 2023-08-28
>
> **"Most efforts in building the new dataset are contributed by humans, which is consuming, non reproducible, and potentially non objective;"**
>
> We appreciate the reviewer's concern regarding the time and resource-intensive nature of human-annotated datasets. While it's true, it remains the *gold standard* in NLP and open-domain QA evaluations for ensuring high-quality data (e.g., TydiQA,  Natural Questions, SQuAD, OK-VQA all rely on human annotation).
> To address the concern of cost, we have also developed an automatically constructed dataset, InfoSeek_Wikidata 1M.
> - **Reproducibility**: We commit to open-sourcing the dataset under the **Apache-2.0 license** upon the EMNLP anonymity period.
> - Quality: We conduct rigorous quality control during data collection with high quality evaluation accuracy at 95% (L259).
>
> By balancing human-annotated and automatically constructed datasets, we aim to offer a comprehensive resource for the community.
>
>
>
> **(Clarification) Q1: I’m not sure whether a dataset can be claimed as automatically generated if no language models are involved in generation.**
>
> We appreciate your observation regarding the term "automatically generated" in the context of our dataset creation. It's true that the recent surge in LLM-based data generation might set a particular expectation for the term. Our aim with this dataset is to maximize scalability while minimizing, but not entirely eliminating, human intervention for quality control. In light of your comment, we will revise the term to "**automatically constructed**" in the camera-ready version to clarify our approach. Thank you for bringing this to our attention.
>
> **(Clarification) Q2: Details of experiment setting of "70.8% OK-VQA questions can be answered by adults without search engines"**
>
> We will include comprehensive details in the appendix for further clarification. Additionally, we invite the reviewer to check some examples from the OK-VQA data browser via the provided link (https://okvqa.allenai.org/browse.html - e.g., "Question: Did he just block or catch the ball?").
>
> Study details:
> - Five adult annotators each annotate 100 examples
> - In total, 500 randomly sampled examples from the OK-VQA training set
> - Annotators are instructed to categorize each example into one of three categories:
> |Category| Percentage|
> |------------|-----------|
> |Questions that can be answered directly by looking at the corresponding image. | 50.8%|
> |Questions that can be answered without looking at the image. | 20%|
> |Questions requiring a Google search for an answer.| 29.2%|
> ||
>
>
> **(Clarification) Q3: What are the potions of unseen entities and unseen questions in two data splits?**
>
> Please find stats in Table 7:
> |Split| Unseen Question | Unseen Entity|
> |-----|-------------------------|--------------------|
> |Val | 18,656 | 54,964 |
> |Test | 98,901 | 249,079|
> |Human | 3,248 | 5,683|
> ||
>
> The design principle of InfoSeek is to evaluate "pre-training knowledge" in the multimodal models (L288-290), which is intentionally aligned with the goal of assessing world knowledge in LLMs through open-domain text QA (e.g., TriviaQA/Natural Question in LLaMA-2 report). In this context, the SOTA models achieving top performance on TriviaQA are LLMs that use few-shot prompting, rather than models that are fine-tuned on training set (https://paperswithcode.com/sota/question-answering-on-triviaqa). Similarly in InfoSeek, the unseen questions/entities serve to rigorously assess the generalization abilities of the multimodal models, preventing model overfitting to the training set.
>
> **(Clarification) Q4: how to promise the answers to their questions can be extracted from these articles? and how many questions are generated for one image?**
>
> We ensure that only questions answerable by the articles are included in our dataset, following the TyDiQA-GoldP annotation guidelines (L 1029-1033).
> The main reason we don't show annotators Wiki articles during the question writing stage is to ensure they have no prior knowledge about the question they ask, thus maintaining the info-seeking nature of the question.
>
> Below, we report additional stats and plan to add in the camera-ready version:
> | Split | Question/Image|
> |-----|-----------------|
> |InfoSeek Wikidata| 1.4|
> |InfoSeek Human|1.0|
> ||
>
> *Reference*
>
> - TriviaQA: A Large Scale Distantly Supervised Challenge Dataset for Reading Comprehension, Joshi et al., 2017 ACL
> - Natural Questions: A Benchmark for Question Answering Research, Kwiatkowski et al., 2019, TACL
> - SQuAD: 100,000+ Questions for Machine Comprehension of Text, Rajpurkar et al., 2016, EMNLP
> - TyDi QA: A Benchmark for Information-Seeking Question Answering in Typologically Diverse Languages, Clark et al., 2020, TACL

---

### Official Review · Reviewer_Kp26 · 2023-08-10

**Soundness:** 4

**Excitement:**

3: Ambivalent: It has merits (e.g., it reports state-of-the-art results, the idea is nice), but there are key weaknesses (e.g., it describes incremental work), and it can significantly benefit from another round of revision. However, I won't object to accepting it if my co-reviewers champion it.

**Missing References:**

The latest multimodal models such as MiniGPT4, LLaVA, mPLUG-Owl etc.

**Paper Topic And Main Contributions:**

The paper proposes VQA dataset that focuses on visual info-seeking questions. The authors evaluate the zero-shot, fine-tuning, and retrieval performance of sota pre-trained models on the dataset.

**Questions For The Authors:**

1. when will the dataset be open source? It has been posted for almost half a year now？
2.  Provide a detailed introduction to the model fine-tuning setting.
3. The scale of the BLIP2 and PALI-X models and the difference in pre-training data are significant, the comparison between these two models unfair.

**Reasons To Accept:**

1. The paper constructs over 1 million visual information-seeking QA pairs using the Wikidata database. Compared to previous works, this dataset covers more fine-grained knowledge and visual understanding, which could potentially accelerate community development.
2. The paper validates the dataset's effectivenessthrough fine-tuning on VLPs.

**Reasons To Reject:**

1. This work has limited innovation, except for proposing a dataset；
2. This  evaluation has too few models，lacks some of the latest multimodal models including SFT such as MiniGPT4, LLaVA, etc.

**Reproducibility:**

3: Could reproduce the results with some difficulty. The settings of parameters are underspecified or subjectively determined; the training/evaluation data are not widely available.

**Reviewer Confidence:**

5: Positive that my evaluation is correct. I read the paper very carefully and I am very familiar with related work.

---

> ### Author Rebuttal · Authors · 2023-08-28
>
> **(Add. Experiment) Q1: Missing reference of "The latest multimodal models such as MiniGPT4, LLaVA, mPLUG-Owl etc."**
>
> We are grateful for Reviewer's expert insight into multimodal models and for highlighting seminal works like MiniGPT4, LLaVA, and mPLUG-Owl. We agree that these models have made significant contributions to the community. In response, we have expanded our evaluation to include a fine-tuned version of mPLUG-Owl on InfoSeek, and the results are included below (we will add them to camera-ready with details of the fine-tuning process).
>
> We feature five multimodal models spanning publications from 2022 to 2023, the most recent of which was on arXiv a month before the EMNLP deadline.
> Our initial selection criteria were primarily guided by models with published SOTA results on VQAv2 or OK-VQA. However, we recognize that the models suggested by the reviewer have a particular emphasis on open-domain conversations, and we are in the process of including them in our evaluation for camera-ready.
>
>
> | Model | arXiv date | report results on VQAv2/OK-VQA? | InfoSeek_Human |
> |----------|---------------|:------------------:|-----------:|
> | OFA (L1192) | 2022/02/07 | Yes |  4.0 |
> | PaLI-17B | 2022/09/14|Yes| 8.2 |
> | BLIP2 | 2023/01/17 |Yes| 9.4|
> | LLaVA | 2023/04/17 |N\A| todo |
> | MiniGPT4 | 2023/04/20 |N\A| todo |
> | mPLUG-owl | 2023/04/27 |N\A| 7.7 |
> | InstructBLIP|2023/05/11|Yes| 9.9 |
> | PaLI-X | 2023/05/29 |Yes| 10.8 |
>
> *- Experiment hypeparameters: LoRA (default) fine-tuning (MAGAer13/mplug-owl-llama-7b-pt) language decoder with lr=2e-4, batch size=8, 60k iterations, on a single A40 GPU.*
>
> ---
>
> **(Clarification) Q2: "when will the dataset be open source?"**
>
> - We are fully committed to facilitating the availability of our dataset and would make the dataset **publicly available under the Apache-2.0 license** (upon the end of EMNLP anonymity period).
>
> ---
>
> **(Clarification) Q3: "The scale of the BLIP2 and PALI-X models and the difference in pre-training data are significant, the comparison between these two models unfair."**
>
> We agree that this is a factor to be considered when reading the experiment, and would emphasize it in our experiment (though we already highlighted the model scale explicitly in Table 4).
>
> However, we want to emphasize that our mission is to help the community understand the current SoTA model’s capability in answering visual infoseek questions, so models are chosen based on their leading multimodal performances rather than pre-training recipes.
>
> MiniGPT4 and LLaVA (which were suggested above) are also not trained at a similar scale, but we agree it would be beneficial to include these experiments, with appropriate caveats highlighted the text and captions.
>
> **(Clarification) Q4: "Provide a detailed introduction to the model fine-tuning setting."**
>
> Detailed instruction on the fine-tuning setting is in Appendix B.1 - "Implementation details of the baseline - B.2 Without-KB models" (L1178) and hyperparameters are summarized in Table 9.

---

### Meta-Review · Area_Chair_FYzc · 2023-09-10

**Recommendation:** 5

**Metareview:**

The paper proposes a dataset on challenging the open-knowledge of VL models. The purpose of this dataset is clear to me, and I like the OKVQA comparisons. As a dataset paper, the paper's method part also nicely elucidates the status of existing approaches, which compares the zero-shot, fine-tuning (i.e., like using model's internal KB), and explicit KB. The overall discussion went positive on the decision of this paper.

---

### Decision · Program_Chairs · 2023-10-07

**Decision:**

Accept-Main

**Comment:**

The paper proposes a dataset on challenging the open-knowledge of VL models. The purpose of this dataset is clear to me, and I like the OKVQA comparisons. As a dataset paper, the paper's method part also nicely elucidates the status of existing approaches, which compares the zero-shot, fine-tuning (i.e., like using model's internal KB), and explicit KB. The overall discussion went positive on the decision of this paper.